# A Comparison of Stress Perception in International and Local First Semester Medical Students Using Psychometric, Psychophysiological, and Humoral Methods

**DOI:** 10.3390/ijerph15122820

**Published:** 2018-12-11

**Authors:** Daniel Huhn, Carolin Schmid, Rebecca Erschens, Florian Junne, Anne Herrmann-Werner, Andreas Möltner, Wolfgang Herzog, Christoph Nikendei

**Affiliations:** 1Department of General Internal Medicine and Psychosomatics, University Hospital Heidelberg, Ruprecht-Karls-University Heidelberg, 69115 Heidelberg, Germany; Carolin_Schmid@gmx.de (C.S.); Wolfgang.Herzog@med.uni-heidelberg.de (W.H.); Christoph.Nikendei@med.uni-heidelberg.de (C.N.); 2Department of Psychosomatic Medicine and Psychotherapy, Medical University Hospital Tubingen, Eberhard-Karls-University Tubingen, 72076 Tubingen, Germany; Rebecca.Erschens@med.uni-tuebingen.de (R.E.); Florian.Junne@med.uni-tuebingen.de (F.J.); Anne.Herrmann-Werner@med.uni-tuebingen.de (A.H.-W.); 3Competence Centre for Examinations in Medicine, Baden-Württemberg, 69120 Heidelberg, Germany; Andreas.Moeltner@med.uni-heidelberg.de

**Keywords:** undergraduate medical education, international students, stress, heart rate variabilities, hair cortisol

## Abstract

(1) Medical doctors and medical students show increased psychological stress levels. International medical students seem to be particularly vulnerable. (2) We compared different methods of assessing stress levels in international and local first year medical students. First, study participants completed questionnaires related to stress, depression, empathy, and self-efficacy (MBI, PSQ, PHQ-9, JSPE-S, and GSE) at three separate points in time (T1 to T3). Second, their heart rate variabilities (HRVs) were recorded in an oral examination, a seminar, and in a relaxing situation. Third, hair samples were collected at the beginning and at the end of the semester to assess the cortisol concentration. (3) Included were 20 international and 20 local first semester medical students. At T1, we found considerable differences between international and local students in the JSPE-S; at T2 in the MBI factor “professional efficacy”, the PHQ-9, and in the JSPE-S; and at T3 in the MBI factors “cynicism” and “professional efficacy”, the PHQ-9, and in the JSPE-S. International and local students also differed concerning their HRVs during relaxation. Over the course of the semester, international students showed changes in the MBI factors “emotional exhaustion” and “professional efficacy”, the PHQ-9, and the GSE. Local students showed changes in the GSE. No effects were found for students’ hair cortisol concentrations. (4) All participants showed low levels of stress. However, while international students experienced their stress levels to decrease over the course of the semester, local students found their stress levels to increase.

## 1. Introduction

The medical profession is associated with a substantial burden that might have an impact on the quality of the physicians’ private life. Furthermore, high levels of stress may affect patient care and lead to a decrease in care quality [1]. Compared to population-based samples, physicians show an increased prevalence of burnout, and this trend is already visible in medical students [2,3]. Burnout is defined as a syndrome consisting of emotional exhaustion (tiredness, somatic symptoms, and decreased emotional resources), depersonalization (i.e., physicians developing negative, cynical attitudes, and impersonal feelings towards their clients), and a lack of feelings of personal accomplishment (feelings of incompetence, inefficiency, and inadequacy) [4,5]. With the beginning of their course of study at university, undergraduate medical students show an increase in psychological stress [6,7], with burnout rates ranging from 28% to 45% [8]. In comparison, resident doctors have burnout prevalence rates ranging from 27% to 75%. These numbers underline the importance of investigating medical students’ stress levels, especially as they impact clinical practice [9].

International students seem to belong to a particularly vulnerable subgroup of all medical students and report higher levels of stress than local students [10,11]. This might be due to the fact that international students studying abroad are confronted with a variety of challenges because of language deficits and cultural barriers [12,13]. Furthermore, international students often complain about a lack of social contacts [14], insufficient support [15], and reduced health-related quality of life [16]. Thus, the prevalence of burnout may be even higher in international medical students. However, not much is known about psychophysiological stress correlates in international medical students in different learning settings.

Heart rate variability (HRV) is a widely used psychophysiological marker for stress. It is a parameter that reflects the heart’s neuro-vegetative activity as well as its autonomous function. It describes the heart’s ability to adapt efficiently to different requirements by altering the time lag from heartbeat to heartbeat depending on the load [17]. Hence, the HRV can be seen as an apt way of measuring of an individual’s adaptability to exogenous and endogenous influences [18]. Over the past 20 years, assessing the HRV has been established as a reliable, non-invasive diagnostic procedure. The clinical interest was sparked when HRVs were found to be a strong and independent predictor of mortality after myocardial infarction [19]. Currently, HRVs are used to determine an individual’s general medical condition as well as their physiological adaptability. Previous studies dealing with HRVs have mainly focussed on patients with specific clinical presentations in different medical sectors [20]. Furthermore, it has been shown that HRVs are reduced in stressful situations [21,22]. However, so far, this means of assessment has not been used to measure differences in stress levels between international and local students of medicine.

Cortisol is often labelled as a “stress hormone”, as it is involved in a variety of regulatory functions: in the central nervous system, it is part of the process of learning, memory consolidation, and regulation of emotions; in the metabolic system, it regulates the storage and utilization of glucose; and in the immune system, it controls the extent of the body’s inflammatory responses and the maturation of lymphocytes [23,24]. These findings have led to a number of theories linking cortisol to stressors, diseases, medical conditions, and lifestyle problems [25]. In a majority of these studies, stress seems to trigger somatic diseases by leading to an increased output of cortisol [26]. However, there are also studies claiming the opposite: a stress-induced suppression of cortisol is made responsible for emerging diseases [27]. According to current data, it is assumed that cortisol deviations in both directions might be potentially harmful, and it depends on other factors whether it is an increased or a reduced cortisol level that leads to disease [25]. The most common method of examining the concentration of cortisol is the use of saliva, blood, or urine. However, these assessments have some methodological limitations that can be overcome if cortisol levels are measured via a hair segment analysis [28,29]: the cortisol that is deposited in human hair is a reliable endogenous biomarker that allows us to retrospectively determine an individual’s stress-level over an extended period of time [30]. So far, this method has not been used to compare the stress-levels of international and local medical students.

Our literature review did not yield any results concerning research conducted with first-term international and local medical students using three different means to measure their stress levels over the course of one semester. The aim of the current study was to compare psychometric, psychophysiological, and humoral stress parameters of first-term international students and first-term local students. We assumed that:

(i) all students would show increased stress-levels in the psychometric assessment;

(ii) these stress-levels would also be detected in the psychophysiological and the humoral measurements; and

(iii) international students would show even higher stress-levels than local students.

## 2. Materials and Methods

The present study is a comparative analysis of different stress parameters in international and local medical students during their first semester at the Medical Faculty of the University of Heidelberg. First-term students were invited to participate in the study at the beginning of the winter term in October 2015. The study was also promoted in Heidelberg’s Tutorial for international Medical students (HeiTiMed) [31]. The first 20 international students and the first 20 local students to answer our invitation via e-mail were included in the study. We examined the following stress parameters: all students completed questionnaires with validated instruments related to stress, fatigue, and depression at the beginning (October 2015), in the middle (December 2015), and at the end of the semester (February 2016). Furthermore, their heart rate variability (HRV) was measured in three different conditions over the course of the semester: during a small seminar, an oral examination, and in a relaxed situation; the latter served as the baseline measurement. Finally, all of the participants provided samples of hair strands at the beginning and at the end of the semester to assess cortisol levels.

We carefully selected and validated the following five questionnaires and asked the participating students to fill them out at the beginning (week 1; T1), in the middle (week 7–8; T2), and at the end of the semester (week 14; T3):

(1) The Maslach-Burnout-Inventory-Student Survey (MBI-SS) is a modified version of the original by Maslach and colleagues [32], developed for students in particular [33]. Overall, it consists of 15 items. High values in “emotional exhaustion” (five items) and “cynicism” (four items) and low values in “professional efficacy” (six items) indicate that there is a risk of a burnout-syndrome. The validity of the student version (MBI-SS) has been demonstrated for German-speaking countries [34].

(2) The Perceived Stress Questionnaire (PSQ) [35] assesses individual stress experience. High values in the questionnaire correlate with high levels of experienced stress. The German version includes 20 items in total, and its validity has been proved. The questions are divided into four different sections with five items each: “worries”, “tension”, “joy”, and “demands” [36].

(3) The depression module (PHQ-9) of the German version of the Patient Health Questionnaire (PHQ-D) [37] comprises nine items, and is applied for the detection of depressive symptoms. Its reliability and validity have been demonstrated sufficiently [38].

(4) The Jefferson Scale of Physician Empathy: Student Version (JSPE-S) [39] is a questionnaire for the self-evaluation of empathy. High values correlate with a high level of empathy. The questionnaire includes 20 items, and has been especially developed for the field of medical education. Good psychometric characteristics of the German version have been demonstrated [40].

(5) The Scale of General Perceived Self-Efficacy (GSE) [41] includes 10 items and measures how optimistically individuals rate their own competence. High values stand for an optimistic self-assessment. The validity of the German version has been proved [42].

For the measurement of heart rate variability (HRV), all students were provided with Polar^®^ V800 heart rate monitors (Polar Electro Ltd., Büttelborn, Germany). The validity of the Polar^®^ V800 and its ability to produce RR (R is a point corresponding to the peak of the QRS complex of the electrocardiography wave; RR is the interval between successive Rs) recordings consistent with an electrocardiograph have been demonstrated [43]. They were asked to use these HRV monitors with the accompanying chest belt sensors in the following three different conditions:an oral examination;a seminar; anda relaxing situation that served as the baseline measurement.

The Polar^®^ V800 heart rate monitors were deployed in five cycles with up to 12 students each in the first semester. International and local students were evenly distributed to these five cycles. Of all the data provided by the students, predefined identical time frames (starting 2 min after the beginning of the measurement) of 15 min each were selected, so that each of the three separate situations (oral examination, seminar, and relaxation) could be compared.

To analyse the obtained data, we examined six different parameters. Three of them (the standard deviation of all NN-(‘NN’ is used instead of RR to emphasize the fact that the processed beats are ‘normal’ beats) intervals (SDNN), the root mean square of successive differences (RMSSD), and the percentage of consecutive NN-intervals that differ more than 50 ms from each other (pNN50)) are time-domain-specific, and the other three (Low-Frequency Power (LF), High-Frequency Power (HF), and the quotient of Low-Frequency Power and High-Frequency Power (LF/HF)) are frequency-domain-specific. The SDNN can be applied as a frequency-independent indicator for the overall variability. It is measured in milliseconds. The RMSSD is the result of taking the square root of the mean of the squares of the successive differences between adjacent NNs. It describes frequency changes between subsequent heart beats which makes it a good parameter for detecting short-term variability and, therefore, an indicator for parasympathetic activity. It is also measured in milliseconds. The parameter pNN50 is measured in per cent, and a high pNN50-value stands for a highly spontaneous change in heart frequency. LF describes a power density spectrum in the frequency range from 0.04 to 0.15 Hertz. Its unit of measurement is square milliseconds. Both sympathetic and parasympathetic factors contribute to this spectrum, whereby the sympathetic influence prevails. HF describes a power density spectrum in the frequency range from 0.15 to 0.40 Hertz. This parameter is also measured in square milliseconds. Only the parasympathetic nervous system has an influence on this spectrum. Finally, LF/HF is considered to be an indicator of equilibrium between the sympathetic (LF) and parasympathetic nervous system (HF). High values of this parameter indicate an increase in sympathetic activity; low values stand for an increase in parasympathetic activity.

In order to analyse humoral stress levels, we took hair samples of all participating students in the first and the last week of the semester. For this purpose, a piece of hair, approximately 3-mm thick, was cut near the hair root from the back of the head. All samples were sent to the biochemical laboratory of the University of Dresden, chair holder Prof. Dr. Clemens Kirschbaum (see, for example, [29]), for the assessment of the hair cortisol concentration. For a more detailed description of the procedures involved in analysing cortisol levels in hair, please refer to the same publication [29].

The ethics committee of Heidelberg University gave their approval for the study design as described above (Number: S-338/2015). The study was conducted in accordance with the declaration of Helsinki (revised form, Fortaleza (Brazil), 2013) [44]. Participation in the study was voluntary, and we obtained the written informed consent of all students prior to their participation. Each participant received 25€ at the end of the semester to cover all expenses.

The data analysis was conducted using the “Statistical Package for the Social Sciences” (SPSS) for Windows, version 22 (IBM Corp., Armonk, NY, USA). As some variables were not normally distributed and variance homogeneity was not given in all cases, we decided to use nonparametric methods. Mann–Whitney U Tests were used to compare group differences in psychometric parameters, psychophysiological values, and hair cortisol concentrations. Friedman tests (in psychometric evaluations: T1 versus T2 versus T3; in psychophysiological measurements: relaxation versus seminar versus exam; in humoral measurements: T1 versus T2) followed by post hoc Dunn–Bonferroni tests were used to assess changes over time. As multiple comparisons for the two groups were performed, all significance levels were adjusted according to Bonferroni. As for effect size, Pearson’s correlation coefficient r was calculated for the comparisons of local and international students and the Dunn–Bonferroni Tests.

In the case of heteroscedasticity, we used the Welch Test to compare group differences between the current sample and representative norm samples or other comparative samples.

## 3. Results

### 3.1. Sample Description

We included the first 20 international and the first 20 local students who responded to our invitation by e-mail in our study. Three international students withdrew their consent to participate in the study at the time when the HRV measurements were to be carried out. Thus, the final sample consisted of 37 medical students in their first semester at the University of Heidelberg; 20 of them were local, and 17 were international. Thirty-six students filled out the questionnaires at all three times of measurement. All students were able to take HRV measurements under the three different conditions, and all students provided hair samples at the beginning and at the end of the semester.

For more details, see Table 1.

### 3.2. Differences between International and Local Students

Due to the small sample size in combination with multiple comparisons and the resulting Bonferroni adjustments, none of the emerging differences between local and international students reached significance. Nevertheless, effect sizes revealed differences between local and international students concerning psychometric results with medium effect sizes (*r* > 0.30) [45]: At T1, local and international differed from each other in the JSPE-S; international students felt less empathetic than local students (*U* = −2.154, *r* = 0.36). At T2, the two groups differed in the MBI factor “professional efficacy”, again in the JSPE-S, and in the PHQ-9. International students’ perceived “professional efficacy” was lower than in local students (*U* = −2.096, *r* = 0.35) and international students also scored lower in the JSPE-S (*U* = −1.921, *r* = 0.32). In the PHQ-9, Local students reported increased feelings of sorrow and depression, while international students scored much lower (*U* = −2.385, *r* = 0.40). Finally, at T3, the two student groups differed in four parameters: the MBI factor “cynicism” (*U* = −2.140, *r* = 0.36), the MBI factor “professional efficacy” (*U* = −2.314, *r* = 0.39), again in the JSPE-S (*U* = −2.549, *r* = 0.42), and again in the PHQ-9 (*U* = −2.665, *r* = 0.44). International students still felt less competent in study-related issues, they also felt less empathetic, and again reported fewer depressive symptoms. Regarding the MBI factor “cynicism”, international students scored substantially lower than their local counterparts. All psychometric variables were also analysed for gender differences, but none emerged (also see Table 2 as well as Figure 1 and Figure 2).

Looking at the psychophysiological results, local and international students differed substantially with regard to the SDNN parameter during relaxation (*U* = −1.707, *r* = 0.28). International students showed a higher variability than local students, indicating that they were more successful in relaxing. We did not find any differences between female and male students regarding their psychophysiological values (also see Table 2 as well as Figure 3).

Concerning the amount of hair cortisol, local and international students did not differ from each over time or due to gender (also see Table 2 as well as Figure 4).

### 3.3. Differences over the Course of the Semester

None of the differences in psychometric variables was significant due to the small sample size in combination with multiple comparisons and the resulting Bonferroni adjustments across the different assessment points. However, the calculated effect sizes revealed four changes in international students across time with almost medium effect sizes: with regard to the MBI factor “emotional exhaustion”, they felt less exhausted over the course of the semester (T1 to T3: *z* = 0.938, *r* = 0.23). Concerning the MBI factor “professional efficacy”, international students showed declining values throughout the semester (T1 to T2: *z* = 1281, *r* = 0.32; T1 to T3: *z* = 1.062, *r* = 0.26), and showed a substantial increase during the semester in the Scale of General Perceived Self-Efficacy (GSE) (T1 to T3: *z* = 0.938, *r* = 0.23). In addition, finally, in the PHQ-9, international students showed a considerable decline in depressive symptoms over the course of the semester (T1 to T2: *z* = 1.031, *r* = 0.26; T1 to T3: *z* = 1.125, *r* = 0.32). In local students, there was one slight change over time: they also showed a slight increase in the GSE scores over the course of the semester (T1 to T3: *z* = −0.800, *r* = 0.18) (also see Table 3 as well as Figure 1 and Figure 2).

Concerning the psychophysiological results, both international and local students showed significant differences within the three conditions (exam, seminar, base line) for all parameters (SDNN, RMSSD, pNN50, LF, HF, and LF/HF), indicating that all the students’ HRVs differed considerably within the three evaluated conditions. The students had the ‘best’ HRV values during relaxation, the values were ‘worse’ during the seminar, and ‘worst’ during the oral examination (also see Table 3 as well as Figure 3).

With regard to hair cortisol concentrations, no considerable changes occurred in international as well as local students over the course of the semester (also see Table 3 as well as Figure 4).

### 3.4. Comparison between Current Sample and Norm Values

For a better interpretation of our results, comparisons between the current student sample and norm groups were performed.

To the best of our knowledge, there is no MBI-SS validation study reporting mean values and standard deviations for a control sample to compare our current study results to [3]. Compared to a German students’ norm sample (*M* = 0.34, *SD* = 0.16), our study’s participants did not differ significantly (T1: *M* = 0.42, *SD* = 0.08; T2: *M* = 0.42, *SD* = 0.09; T3: *M* = 0.42, *SD* = 0.11; *t*(280) = 4.76, *p* > 0.05) in the PSQ [36]. With respect to the PHQ-9, students in our sample presented significantly higher scores at T1 (*M* = 4.38, *SD* = 3.26) than a German comparative sample (*M* = 3.56, *SD* = 4.08; *t*(2097) = 1.69, *p* < 0.05) [46]. However, no significant differences compared to the comparative sample emerged (*t*(2097) = −0.22, *p* > 0.05) at the other two points of measurement (T2: *M* = 3.51, *SD* = 3.20; T3: *M* = 3.33, *SD* = 3.22). Concerning the JSPE-S, no significant differences emerged between students in the current study (T1: *M* = 107.3, *SD* = 12.9; T2: *M* = 106.2, *SD* = 14.4; T3: *M* = 105.8, *SD* = 13.8) and a comparative sample of Austrian students (*M* = 110.5, *SD* = 12.5; *t*(550) = −1.99, *p* > 0.05) [47]. In the GSE, students did not differ significantly at the first two points of measurement (T1: *M* = 30.07, *SD* = 3.36; T2: *M* = 30.59, *SD* = 4.10) from a representative German sample (*M* = 29.43, *SD* = 5.36; *t*(2053) = 1.69, *p* > 0.05) [42]. However, at T3, students in our study presented significantly higher values (*M* = 32.06, *SD* = 4.01; *t*(2053) = 3.87, *p* < 0.05).

All students’ presented HRVs were within a normal range. Students’ RMSSD values (resting condition: *M* = 59.8; seminar: *M* = 43.3; exam: *M* = 14.7) were significantly higher than values from age-matched healthy controls (resting: *M* = 14.2; deep-breathing: *M* = 23.8) [48], which means they were ‘better’ and can, therefore, be considered to be ‘normal’.

With regard to the amount of hair cortisol, students in our study had significantly higher values (T1: *M* = 5.89, *SD* = 5.77; T2: *M* = 5.41, *SD* = 4.18) than in comparable healthy samples [49].

## 4. Discussion

The current study is the first to compare psychometric, psychophysiological, and humoral assessments of stress perception between international and local first-term medical students. In contrast to other studies [50,51], our study’s participants did not present significantly higher levels of stress with the exception of two measures: first, our sample’s students reported higher levels of depression in the PHQ-9 questionnaire at the beginning of the semester. However, as their scores only reached the cut-off values equivalent to ”minimal” depression symptoms, they should not be overrated [37]. Second, students in our sample showed significantly higher quantities of hair cortisol than comparable samples. However, we are unable to interpret these findings, as our small sample size does not allow us to control for parameters, such as sex, age, and ethnicity, which may potentially affect hair cortisol measurements [49]. International and local students differed clearly from each other regarding most of the psychometrical parameters as well as their heart rate variabilities during relaxation. Unfortunately, none of these differences were significant. This can be explained by the small sample size and Bonferroni correction after multiple comparison testing. No considerable differences emerged in regard to international and local students’ hair cortisol concentrations. Our analysis revealed a similar direction concerning the psychometric items “emotional exhaustion”, “cynicism”, and “depression”. International students showed moderate stress-levels at the beginning of their course of study, which decreased during the semester, whereas local students presented very low stress-levels at the beginning of the semester, which increased during the semester. At the end of the semester, international students showed lower psychometric scores for stress than their local counterparts.

In general, the beginning of a course of study could be regarded as a kind of threshold for young people. Interestingly, our findings indicate that this phase might be a completely different experience for international and local students. International students have to start planning their studies abroad for a longer time in advance and have several administrative hurdles to overcome. They are expected to attend preparation courses and provide language certificates. Furthermore, they have to start looking for accommodation, organise their student visa, and make travel arrangements. All these aspects can be very challenging for young international applicants and can contribute to stress [52]. The greater the cultural difference between the home culture and the new culture is perceived, the more difficult the process of acculturation will be [53]. Local students’ preparations prior to starting university may also be stressful, but on a different level. After their high school degree, local students apply for a place to study. This might be in another German town or city, or at the university nearest to their home. Local students are not confronted with linguistic or intercultural challenges and sometimes even remain in their social environments. Thus, it is not surprising that our study indicated that international and local students have different levels of self-esteem and diverging opinions of themselves.

Regarding self-efficacy in study-related issues, international students showed decreasing feelings of competence and successful achievement over the course of the semester. In contrast, local students only slightly changed in their self-evaluation. Other studies have revealed that international medical students perform poorer than local students in written, oral, and practical examinations, especially during the first years of their studies [54,55,56,57,58]. Furthermore, international students often have an extended time of study [59] as well as higher drop-out rates [54,60]. Mostly, the weaker academic performance is ascribed to linguistic difficulties and cultural barriers, rather than to factors related to the subject of studies [12,13]. However, the fact that it is harder for international students to do well at university seems to make them feel less competitive than their local counterparts, even though they already had to give an extraordinary performance to receive a place to study at a German university in the first place.

Nevertheless, concerning their experienced self-efficacy in general, both international and local students presented very similar scores that increased over the three times of measurement. It can be assumed that, even though international students had less confidence in their own professional capabilities, they were still optimistic of their overall competence concerning future situations. This might be explained by the fact that the Scale of General Perceived Self-Efficacy does not measure study-related issues, but rates a general feeling of self-effectiveness. Concerning “empathy”, all the students presented relatively high scores that remained unchanged over the course of the semester. However, international students showed a clearly lower score than local students. Instead of explaining these findings with the different cultural backgrounds, it seems more plausible to discuss other aspects: Studies have indicated that medical students may show a decline in empathy if they are exposed to a high workload, a reduced quality of life, distress, reduced contact with their family, and a lack of social support from peer groups [61].

In contrast to the discussed differences concerning “emotional exhaustion”, there was no difference between international and local students regarding the scores of the Perceived Stress Questionnaire. We could interpret this result as being caused by the fact that the item “emotional exhaustion” is focussed specifically on work- or study-related stress, whereas the Perceived Stress Questionnaire refers to stress more generally. The participating students seemed to experience themselves as more stressed by study-related topics, which fortunately did not affect other areas of life.

The students’ heart rate variabilities differed considerably over the three conditions we examined. While heart rate variabilities showed the highest values during relaxation in all relevant domains, these scores declined from the seminar to the examination condition, indicating that the students’ inner tension increased from relaxation to the exam situation. These results are in line with previous studies that indicate a decreased value of the overall heart rate variability and high-frequency components as well as an increased normalized low frequency under stressful conditions, such as an oral exam [17,62]. No differences between international and local students could be detected within the different conditions that were assessed. However, the three different conditions and the students’ nationalities (international or local) were mutually dependent on each other: concerning the standard deviation of all NN intervals, international and local students revealed clear differences depending on the three different conditions. While international and local students both presented the same heart rate variability during the seminar and the oral examination, their values differed considerably during relaxation. Here, local students displayed even lower variabilities than in the seminars, while international students presented higher variabilities. These findings indicate that international students were more able to self-regulate during relaxation. Local students, on the other hand, showed the same level of arousal during relaxation and while being at university. A possible explanation could be that international students have become used to stressful situations due to their experiences before starting university and are, therefore, more able to unwind in relaxing situations.

Concerning hair cortisol concentrations, we did not detect any differences between international and local students. An important note is that all cortisol concentrations were within the normal range and comparable to those of healthy control groups from other studies [63].

### Limitations

The current study has several limitations: first, the low sample size has to be taken into account. Second, the students themselves were responsible for the psychophysiological measurements. Although they were briefed in advance about how to start and stop the heart rate monitors and record the data afterwards, the measurements were not supervised as one of the measuring conditions was an oral exam. Third, the participation in the study was not randomized. All first-year students were informed about the research project, and we included the first 20 local and 20 international students to respond by e-mail. Fourth, we had three dropouts in the international group, but none in the local group. Therefore, we cannot rule out a systematic error. As all dropouts occurred at an advanced stage of the study, we were unable to include further students.

## 5. Conclusions

In summary, our sample of first-term medical students showed low levels of stress. However, there were some differences between international and local students. Assessed with five different questionnaires, the psychometrical parameters can be regarded as the students’ experienced stress at the clinical level. The international students experienced their stress levels to decrease over the course of the semester, while the local students found their stress levels to increase. Despite this divergent trend of results, it has to be emphasized that the stress levels were low at all points of measurement. The psychometric results revealed that the international students perceived themselves to be less competent concerning their academic achievements and presented lower levels of empathy than local students. The HRV measurement results were regarded as a parameter for stress levels in changing conditions. Our results found some differences between the international and local students: during relaxation, the international students presented much higher variabilities than the local students. During the seminar and oral exam, the HRVs of international and local students did not differ from each other. Finally, the hair cortisol concentration was assessed as a long-term marker for stress. Here, all students displayed low values, indicating that they were not exposed to harmful stress. Even though the comparison of hair cortisol concentration of international and local students did not yield significant results, it was thought-provoking to note that international students showed lower cortisol levels.

## Figures and Tables

**Figure 1 ijerph-15-02820-f001:**
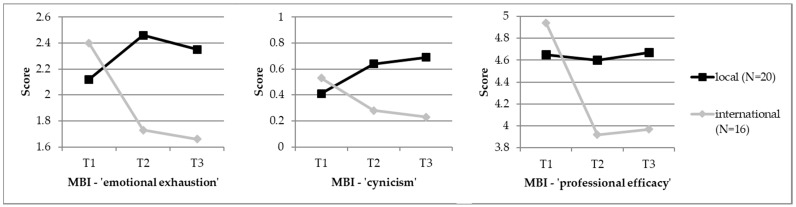
Line charts depicting differences between international and local students concerning the three MBI factors ‘emotional exhaustion’, ‘cynicism’, and ‘professional efficacy’.

**Figure 2 ijerph-15-02820-f002:**
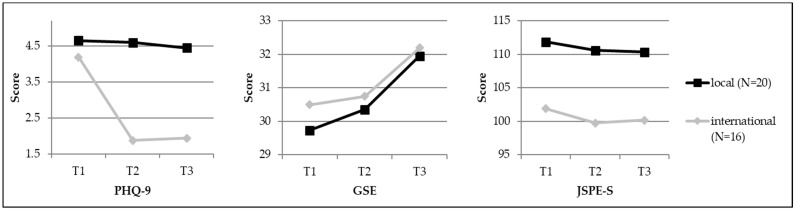
Line charts depicting differences between international and local students concerning the ‘PHQ-9’, the ‘GSE’, and the ‘JSPE-S’.

**Figure 3 ijerph-15-02820-f003:**
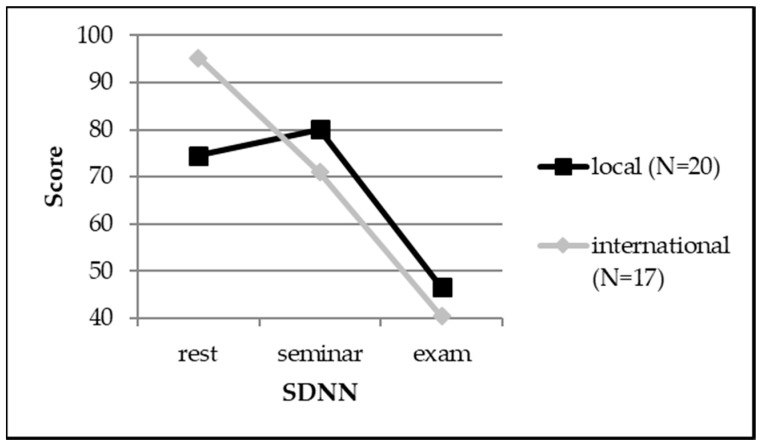
A line chart depicting differences between international and local students concerning the SDNN of heart rate variabilities (HRVs).

**Figure 4 ijerph-15-02820-f004:**
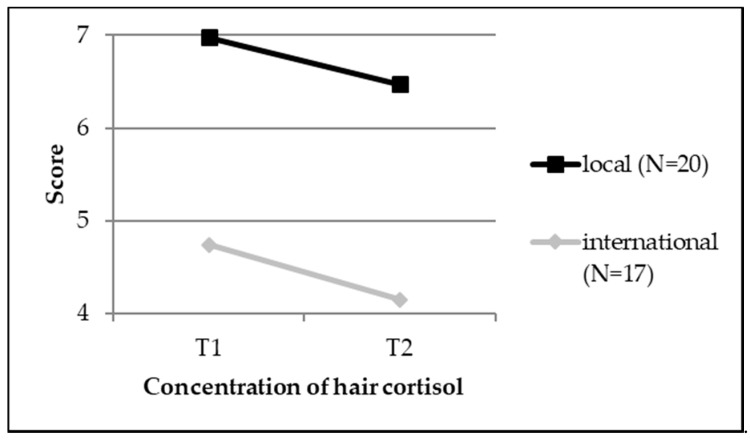
A line chart depicting differences between international and local students concerning the concentration of hair cortisol.

**Table 1 ijerph-15-02820-t001:** The descriptive presentation of the sample.

Students’ Cultural Background	*N*	Male	Female	*M* _age_	*SD* _age_
Local	20	7	13	19.80	2.12
International	17	7	10	20.67	3.94
All	37	14	23	20.21	3.10
**International Students’ Origin**					
Europe	6				
Southeast Asia	4				
Latin America	3				
Middle East	2				
Africa	1				
South Asia	1				

**Table 2 ijerph-15-02820-t002:** Mann–Whitney U Tests for differences in local and international students.

**Psychometric Measure**	**Nationality**	***N***	***M_T1_***	***SD_T1_***	***Mean Rank***	***Rank-Sum***	***U***	***r***	***M_T2_***	***SD_T2_***	***Mean Rank***	***Rank-Sum***	***U***	***r***	***M_T3_***	***SD_T3_***	***Mean Rank***	***Rank-Sum***	***U***	***r***
*MBI, ‘emotional exhaustion’*	local	20	2.12	0.93	17.60	352.0	−0.856	0.14	2.46	1.02	22.08	441.5	−1.510	0.25	2.35	1.20	21.00	420.0	−1.596	0.27
	international	16	2.40	1.00	20.65	351.0			1.73	1.05	16.64	299.5			1.66	1.04	15.38	246.0		
*MBI*, *‘cynicism’*	local	20	0.41	0.73	19.13	382.5	−0.087	0.01	0.64	0.95	21.00	420.0	−0.930	0.16	0.69	0.89	21.60	432.0	−2.140	0.36
	international	16	0.53	0.93	18.85	320.5			0.28	0.54	17.83	321.0			0.23	0.49	14.63	234.0		
*MBI*, *‘professional efficacy’*	local	20	4.65	0.71	16.75	335.0	−1.378	0.23	4.60	0.71	23.08	461.5	−2.096	0.35	4.67	0.86	22.13	442.5	−2.314	0.39
	international	16	4.94	0.61	21.65	368.0			3.92	1.07	15.53	279.5			3.97	0.92	13.97	223.5		
*PSQ-20*	local	20	0.42	0.08	17.43	348.5	−0.962	0.16	0.42	0.72	19.95	399.0	−0.582	0.10	0.43	0.13	18.85	377.0	−0.224	0.04
	international	16	0.44	0.11	20.85	354.5			0.41	0.12	17.88	304.0			0.41	0.08	18.06	289.0		
*JSPE-S*	local	20	111.85	10.19	22.53	450.5	−2.154	0.36	110.60	11.34	22.15	443.0	−1.921	0.32	110.35	10.77	22.50	450.0	−2.549	0.42
	international	16	101.86	14.51	14.85	252.5			99.75	15.81	15.29	260.0			100.19	15.37	13.50	216.0		
*PHQ-9*	local	20	4.65	3.54	19.90	398.0	−0.553	0.09	4.60	3.17	22.88	457.5	−2.385	0.40	4.45	3.20	22.63	452.5	−2.665	0.44
	international	16	4.19	3.02	17.94	305.0			1.88	2.47	14.44	245.5			1.94	2.72	13.34	213.5		
*GSE*	local	20	29.73	3.35	17.68	353.5	−0.812	0.14	30.35	4.45	18.28	365.5	−0.444	0.07	31.95	4.11	18.98	379.5	−0.304	0.05
	international	16	30.50	3.54	20.56	349.5			30.75	3.84	19.85	337.5			32.19	4.02	17.91	286.5		
**Psychophysiological measure**	**Nationality**	***N***	***M_rest_***	***SD_rest_***	***Mean Rank***	***Rank-Sum***	***U***	***r***	***M_seminar_***	***SD_seminar_***	***Mean Rank***	***Rank-Sum***	***U***	***r***	***M_exam_***	***SD_exam_***	***Mean Rank***	***Rank-Sum***	***U***	***r***
*SDNN (ms)*	local	20	74.56	26.24	16.20	324.0	−1.707	0.28	80.13	30.86	20.70	414.0	−1.036	0.17	46.58	19.94	20.25	405.0	−0.762	0.13
	international	17	95.20	40.37	22.29	379.0			70.94	28.52	17.00	289.0			40.46	17.15	17.53	298.0		
*RMSSD (ms)*	local	20	59.45	35.31	18.40	368.0	−0.366	0.06	47.21	29.68	19.90	398.0	−0.549	0.09	15.47	9.25	19.75	395.0	−0.457	0.08
	international	17	60.25	29.56	19.71	335.0			38.64	17.74	17.94	305.0			13.72	7.61	18.12	308.0		
*pNN50 (%)*	local	20	30.85	24.64	18.30	366.0	−0.427	0.07	21.95	20.23	19.90	398.0	−0.549	0.09	2.28	2.83	20.35	407.0	−0.823	0.14
	international	17	31.58	19.69	19.82	337.0			16.31	12.77	17.94	305.0			1.65	2.14	17.41	296.0		
*LF (ms²)*	local	20	1716	1409	17.75	355.0	−0.762	0.13	1784	1250	19.15	383.0	−0.091	0.02	840.0	660.1	20.35	407.0	−0.823	0.14
	international	17	1932	1409	20.47	348.0			1782	1636	18.82	320.0			654.8	644.5	17.41	296.0		
*HF (ms²)*	local	20	1657	1941	17.70	354.0	−0.792	0.13	1163	1509	18.75	375.0	−0.152	0.03	165.1	193.7	19.80	396.0	−0.488	0.08
	international	17	1655	1582	20.53	349.0			785	866	19.29	328.0			136.3	138.6	18.06	307.0		
*LF/HF*	local	20	2.28	1.93	19.20	384.0	−0.122	0.02	3.62	2.37	20.45	409.0	−0.884	0.15	7.52	5.16	19.95	399.0	−0.579	0.10
	international	17	2.04	1.64	18.76	319.0			3.02	2.14	17.29	294.0			6.21	2.84	17.88	304.0		
**Humoral measure**	**Nationality**	***N***	***M_T1_***	***SD_T1_***	***Mean Rank***	***Rank-Sum***	***U***	***r***	***M_T2_***	***SD_T2_***	***Mean Rank***	***Rank-Sum***	***U***	***r***						
*Hair cortisol (pg/mg)*	local	20	6.98	7.53	20.50	410.0	−0.585	0.10	6.47	5.12	21.25	425.0	−1.371	0.23						
	international	17	4.74	2.53	18.39	331.0			4.15	2.27	16.35	278.0								

SDNN, the standard deviation of all NN-intervals; RMSSD, the root mean square of successive differences; pNN50, the percentage of consecutive NN-intervals that differ more than 50 ms from each other; LF, Low-Frequency Power; HF, High-Frequency Power; LF/HF, the quotient of Low-Frequency Power and High-Frequency Power.

**Table 3 ijerph-15-02820-t003:** The Friedman Test for changes over the course of the semester.

**Psychometric Measure**	**Nationality**	***N***	***M_T1_***	***SD_T1_***	***Mean Rank***	***M_T2_***	***SD_T2_***	***Mean Rank***	***M_T3_***	***SD_T3_***	***Mean Rank***	***df***	***Chi^2^***
*MBI, ‘emotional exhaustion’*	local	20	2.12	0.93	1.93	2.46	1.02	2.18	2.35	1.20	1.90	2	0.97
	international	16	2.40	1.00	2.59	1.73	1.05	1.75	1.66	1.04	1.66	2	9.56
*MBI, ‘cynicism’*	local	20	0.41	0.73	1.68	0.64	0.95	2.05	0.69	0.89	2.28	2	5.65
	international	16	0.53	0.93	2.16	0.28	0.54	1.84	0.23	0.49	2.00	2	1.67
*MBI, ‘professional efficacy’*	local	20	4.65	0.71	2.03	4.60	0.71	1.80	4.67	0.86	2.18	2	1.54
	international	16	4.94	0.61	2.78	3.92	1.07	1.50	3.97	0.92	1.72	2	16.03
*PSQ-20*	local	20	0.42	0.08	1.95	0.42	0.12	2.03	0.43	0.13	2.03	2	0.076
	international	16	0.44	0.11	2.31	0.41	0.12	1.91	0.41	0.08	1.78	2	2.59
*JSPE-S*	local	20	111.85	10.19	2.00	110.60	11.34	2.08	110.35	10.77	1.93	2	0.25
	international	16	101.86	14.51	2.19	99.75	15.81	1.94	100.19	15.37	1.88	2	0.90
*PHQ-9*	local	20	4.65	3.54	1.95	4.60	3.17	2.05	4.45	3.20	2.00	2	0.12
	international	16	4.19	3.02	2.72	1.88	2.47	1.69	1.94	2.72	1.59	2	14.25
*GSE*	local	20	29.73	3.35	1.65	30.35	4.45	1.90	31.95	4.11	2.45	2	8.12
	international	16	30.50	3.54	1.63	30.75	3.84	1.81	32.19	4.02	2.56	2	9.00
**Psychophysiological Measure**	**Nationality**	***N***	***M_rest_***	***SD_rest_***	***Mean Rank***	***M_seminar_***	***SD_seminar_***	***Mean Rank***	***M_exam_***	***SD_exam_***	***Mean Rank***	***df***	***Chi^2^***
*SDNN (ms)*	local	20	74.56	26.24	2.25	80.13	30.86	2.45	46.58	19.94	1.30	2	15.10 *
	international	17	95.20	40.37	2.76	70.94	28.52	2.00	40.46	17.15	1.24	2	19.88 *
*RMSSD (ms)*	local	20	59.45	35.31	2.65	47.21	29.68	2.30	15.47	9.25	1.05	2	28.30 *
	international	17	60.25	29.56	2.82	38.64	17.74	2.12	13.72	7.61	1.06	2	26.82 *
*pNN50 (%)*	local	20	30.85	24.64	2.75	21.95	20.23	2.15	2.28	2.83	1.10	2	27.90 *
	international	17	31.58	19.69	2.71	16.31	12.77	2.12	1.65	2.14	1.18	2	20.24 *
*LF (ms²)*	local	20	1716	1409	2.35	1784	1250	2.30	840.0	660.1	1.35	2	12.70 *
	international	17	1932	1409	2.53	1782	1636	2.24	654.8	644.5	1.24	2	15.65 *
*HF (ms²)*	local	20	1657	1941	2.65	1163	1509	2.30	165.1	193.7	1.05	2	28.30 *
	international	17	1655	1582	2.76	785	866	2.06	136.3	138.6	1.18	2	21.53 *
*LF/HF*	local	20	2.28	1.93	1.30	3.62	2.37	1.90	7.52	5.16	2.80	2	22.80 *
	international	17	2.04	1.64	1.24	3.02	2.14	1.82	6.21	2.84	2.94	2	25.53 *
**Humoral measure**	**Nationality**	***N***	***M_T1_***	***SD_T1_***	***Mean Rank***	***M_T2_***	***SD_T2_***	***Mean Rank***				***df***	***Chi^2^***
*Hair cortisol (pg/mg)*	local	20	6.98	7.53	1.50	6.47	5.12	1.50				1	0.00
	international	17	4.74	2.53	1.59	4.15	2.27	1.41				1	0.53

* *p* < 0.05.

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
