# Peer review of "A Comparison of Stress Perception in International and Local First Semester Medical Students Using Psychometric, Psychophysiological, and Humoral Methods"

_ijerph, 2018, doi:10.3390/ijerph15122820_

Round 1
Reviewer 1 Report
Huhn et al. A comparison of stress perception…
The additional stress caused by exposure to a foreign culture and language in addition to strenuous studies has been endured by many of us including the present reviewer. It is also compounded by several additional factors including the economic strain it means to the family at home and their hopes and expectations for success.
The study is well conceived as it uses three separate paradigms for studying stress, but the selection of participants, loss of subjects in only one of the groups, data analysis and the verbose language mar an otherwise interesting study.
1. The background, research paradigm and methods of the study are appropriately described. The authors also provide a description of the limitations of the study including the limited number of participants and lack of randomisation. The limitation section should also mention and discuss the confounding effect of the three dropouts in the international group and why new students were not included despite the fact that the samples were non-randomised “convenient” samples.
2. Multivariate statistics (ANOVA/GLM?) are appropriately used to analyse data. However, the method for multiple significance testing is not detailed.
3. The mean and standard deviation are used to express central tendency and variation, respectively. Unfortunately there is no mention of the use of tests for outliers or tests for normal distribution. The authors will understand that multivariate statistical methods are vulnerable to the presence of outliers and to non-normal distributions. In numerous instances, e.g. Table 2, PHQ-9and table 4 2*SD (should normally include 95% of the observations) is larger than the mean itself, strongly indicating skew distribution. The authors need to do a proper exploratory data analysis of all data including tests for outliers and tests for normality and use transformations of the data to normality as appropriate when appropriate.
4. It is clearly stated that the laboratory of Dr. Kirschbaum was engaged to measure the concentrations of cortisol in hair extracts. The preanalytical procedures, especially the techniques for sampling hair are not appropriately described. Since hair grows at in the order of 10 mm/month, it is crucial to know how far from the hair root the samples were taken in order to correlate the concentrations of cortisol in the hair extracts to the time period and thereby to events in the lives of the students in the time period when that part of their hair was sampled.
5. The statement on page 2, line 84 that measurements of cortisol concentrations in hair has not been used to measure stress in international medical studies is simply not true as the already five years old publication will show Faresjo, A., et al. (2013). "Higher Perceived Stress but Lower Cortisol Levels Found among Young Greek Adults Living in a Stressful Social Environment in Comparison with Swedish Young Adults." PLoS One 8(9): e73828.
6. The manuscript is written in correct English which, unfortunately, is unnecessarily verbose. Furthermore, words and concepts from colloquial English as used in the manuscript are not appropriate for describing and discussion scientific results. E.g. the concept “effective” (first used on page 11 line 292 and subsequently) can mean both the ability of the student to get certain study results, or – alternatively how much effort the student needs to obtain certain study results. The authors need to explain carefully what they mean when they are using colloquial English or refer to well-defined scientific concepts.
Author Response
Reviewer 1:
The additional stress caused by exposure to a foreign culture and language in addition to strenuous studies has been endured by many of us including the present reviewer. It is also compounded by several additional factors including the economic strain it means to the family at home and their hopes and expectations for success.
The study is well conceived as it uses three separate paradigms for studying stress, but the selection of participants, loss of subjects in only one of the groups, data analysis and the verbose language mar an otherwise interesting study.
Dear Reviewer, thank you for your positive and personal feedback. We highly appreciate the effort which you have invested in our manuscript. We have tried to address all of your suggestions (see answers to your comments below) and hope that the manuscript has substantially improved in the process.
1. The background, research paradigm and methods of the study are appropriately described. The authors also provide a description of the limitations of the study including the limited number of participants and lack of randomisation. The limitation section should also mention and discuss the confounding effect of the three dropouts in the international group and why new students were not included despite the fact that the samples were non-randomised “convenient” samples.
Thank you for this suggestion. We have now added some further information to the limitation section (page 16, lines 440-442):
“Fourth, we had three dropouts in the international group but none in the local group. Therefore, we cannot rule out a systematic error. As all dropouts occurred at an advanced stage of the study, we were unable to include further students.”
2. Multivariate statistics (ANOVA/GLM?) are appropriately used to analyse data. However, the method for multiple significance testing is not detailed.
Thank you for this remark. We agree that an adjustment of significance levels is needed in our case due to multiple testing. We have accordingly adjusted our new, non-parametric results (see next point) according to Bonferroni. Unfortunately, none of the emerging differences stayed significant which is owed to our small sample size. However, we have now calculated Pearson's correlation coefficient (Pearson's r) as a measure of effect size and our data shows small to medium effects for most of the analysed variables. Hence, we have revised this section of our manuscript very thoroughly and have included the following points in the manuscript:
“As multiple comparisons for the two groups were performed, all significance levels were adjusted according to Bonferroni.” (pages 4-5, lines 198-199)
“Unfortunately, none of these differences were significant. This can be explained by the small sample size and Bonferroni correction after multiple comparison testing.” (page 14, lines 341-343)
3. The mean and standard deviation are used to express central tendency and variation, respectively. Unfortunately there is no mention of the use of tests for outliers or tests for normal distribution. The authors will understand that multivariate statistical methods are vulnerable to the presence of outliers and to non-normal distributions. In numerous instances, e.g. Table 2, PHQ-9and table 4 2*SD (should normally include 95% of the observations) is larger than the mean itself, strongly indicating skew distribution. The authors need to do a proper exploratory data analysis of all data including tests for outliers and tests for normality and use transformations of the data to normality as appropriate when appropriate.
Thank you for this important note. We thoroughly agree that these statistical issues are important and have consequently reanalysed our data. As not all of our variables were normally distributed and variances were not homogeneous in all cases, we decided to reanalyse the data using non-parametric methods. Accordingly, we used the Mann-Whitney-U-Tests as well as the Friedman-Tests. Please find our detailed changes to the manuscript below:
“As some variables were not normally distributed and variance homogeneity was not given in all cases, we decided to use non-parametric methods. Mann-Whitney-U-Tests were used to compare group differences in psychometric parameters, psychophysiological values, and hair cortisol concentrations. Friedman tests (in psychometric evaluations: T1 vs. T2 vs. T3; in psychophysiological measurements: relaxation vs. seminar vs. exam; in humoral measurements: T1 vs. T2) followed by post hoc Dunn-Bonferroni tests were used to assess changes over time. To account for multiple testing, we adjusted all significance levels to Bonferroni. As effect size, Pearson’s correlation coefficient r was calculated for the comparisons of local and international students and the Dunn-Bonferroni-Tests.” (pages 4-5, lines 192-201)
“3.2. Differences between international and local students
Due to the small sample size in combination with multiple comparisons and the resulting Bonferroni adjustments, none of the emerging differences between local and international students reached significance. Nevertheless, effect sizes revealed differences between local and international students concerning psychometric results with medium effect sizes (r > .30) [45]: At T1, local and international differed from each other in the JSPE-S; international students felt less empathetic than local students (U=-2.154, r=.36). At T2, the two groups differed in the MBI-factor “professional efficacy”, again in the JSPE-S and in the PHQ-9. International students’ perceived “professional efficacy” was lower than in local students (U=-2.096, r=.35) and international students also scored lower (U=-1.921, r=.32) in the JSPE-S and in the PHQ-9. Local students reported increased feelings of sorrow and depression, while international students scored much lower (U=-2.385, r=.40). Finally, at T3, the two student groups differed in four parameters: MBI-factor “cynicism” (U=-2.140, r=.36), MBI-factor “professional efficacy” (U=-2.314, r=.39), again in the JSPE-S (U=--2.549, r=.42), and again in the PHQ-9 (U=-2.665, r=.44). International students still felt less competent in study-related issues, they also felt less empathetic, and again reported fewer depressive symptoms. Regarding the MBI-factor “cynicism”, international students scored substantially lower than their local counterparts. All psychometric variables were also analysed for gender differences but none emerged (also see Table 2 as well as Figures 1 and 2).
Looking at psychophysiological results, local and international students differed substantially with regard to the SDNN parameter during relaxation (U=-1.707, r=.28). International students showed a higher variability than local students, indicating that they were more successful in relaxing. We did not find any differences between female and male students regarding their psychophysiological values (also see table 2 as well as figure 3).
Concerning the amount of hair cortisol, local and international students did not differ from each over time or due to gender (also see table 2 as well as figure 4).” (pages 5-6, lines 214-238)
“3.3 Differences over the course of the semester
None of the differences in psychometric variables was significant due to small sample size in combination with multiple comparisons and the resulting Bonferroni adjustments across the different assessment points. However, calculated effect sizes revealed four changes in international students across time with almost medium effect sizes: with regard to the MBI-factor “emotional exhaustion”, they felt less exhausted over the course of the semester (T1 to T3: z=.938, r=.23). Concerning the MBI-factor “professional efficacy”, international students showed declining values throughout the semester (T1 to T2: z=1281, r=.32; T1 to T3: z=1.062, r=.26) and showed a substantial increase during the semester in the Scale of General Perceived Self-Efficacy (GSE) (T1 to T3: z=.938, r=.23). And finally, in the PHQ-9, international students showed a considerable decline in depressive symptoms over the course of the semester (T1 to T2: z=1.031, r=.26; T1 to T3: z=1.125, r=.32). In local students there was one slight change over time: they also showed a slight increase in the GSE scores over the course of the semester (T1 to T3: z=-.800, r=.18) (also see table 3 as well as figures 1 and 2).
Concerning psychophysiological results, both international and local students showed significant differences within the three conditions (exam, seminar, base line) for all parameters (SDNN, RMSSD, pNN50, LF, HF, and LF/HF), indicating that all the students’ HRVs differed considerably within the three evaluated conditions. The students had the ‘best’ HRV-values during relaxation, the values were ‘worse’ during the seminar and ‘worst’ during the oral examination (also see Table 3 as well as Figure 3).
With regard to hair cortisol concentrations, no considerable changes occurred in international as well as local students over the course of the semester (also see Table 3 as well as Figure 4).” (page 6, lines 239-259)
4. It is clearly stated that the laboratory of Dr. Kirschbaum was engaged to measure the concentrations of cortisol in hair extracts. The preanalytical procedures, especially the techniques for sampling hair are not appropriately described. Since hair grows at in the order of 10 mm/month, it is crucial to know how far from the hair root the samples were taken in order to correlate the concentrations of cortisol in the hair extracts to the time period and thereby to events in the lives of the students in the time period when that part of their hair was sampled.
Thank you for this remark. We entirely agree that this information is important to interpret the given results. To make this clearer, we have added the following sentence to the methods section (page 4, lines 175-176):
“For this purpose, a piece of hair, approximately 3-mm thick, was cut near the hair root from the back of the head.”
5. The statement on page 2, line 84 that measurements of cortisol concentrations in hair has not been used to measure stress in international medical studies is simply not true as the already five years old publication will show Faresjo, A., et al. (2013). "Higher Perceived Stress but Lower Cortisol Levels Found among Young Greek Adults Living in a Stressful Social Environment in Comparison with Swedish Young Adults." PLoS One 8(9): e73828.
Thank you for this feedback. We are sorry to have not addressed the above named publication. However, we have now read it with great interest and accordingly changed the information given in the introduction as follows (page 2, lines 94-95):
“So far, this method has not been used to compare stress-levels of international and local medical students.”
6. The manuscript is written in correct English which, unfortunately, is unnecessarily verbose. Furthermore, words and concepts from colloquial English as used in the manuscript are not appropriate for describing and discussion scientific results. E.g. the concept “effective” (first used on page 11 line 292 and subsequently) can mean both the ability of the student to get certain study results, or – alternatively how much effort the student needs to obtain certain study results. The authors need to explain carefully what they mean when they are using colloquial English or refer to well-defined scientific concepts.
Thank you for your valuable appraisal. We agree that the use of the word “effective” is not appropriate in this context and have changed the following two sections accordingly:
“Regarding self-efficacy in study-related issues, international students showed decreasing feelings of competence and successful achievement over the course of the semester.” (page 15, lines 370-372)
“Psychometric results revealed that the international students perceived themselves as less competent concerning their academic achievements and presented lower levels of empathy than local students.” (page 16, lines 451-453)
Furthermore, we have revised the language throughout the manuscript, largely focusing our efforts on improving the discussion’s wording. Please find these changes in the track-change-document.
Reviewer 2 Report
General comment
This research investigated differences in stress between international and local (domestic) students. Five subjective stress scales (MBI, PSQ, PHQ-9, JSPE-S, GSE) and two physiological stress markers (HRV, hair cortisol) were used for stress measurement. The effect of nationality on JSPE-S, PHQ-9 and MBI was observed. The international students showed higher SDNN at rest, however there was no difference during seminar and during examination. Hair cortisol concentration was higher for local (domestic) students than domestic students, however the difference was not statistically significant.
Specific comments
1. Results on hair cortisol should be added to Abstract even if there is no significant difference.
2. The authors should describe more about participant characteristics. Difficulty of studying in Germany might be different between a student from Austria and a student from Japan.
3. Fig. 1-4 are not necessary because the results were numerically demonstrated in the table.
4. What was the basis for concluding that the stress of the students was low? The authors should clarify the grounds for this conclusion. Quantitative comparisons with other previous results are needed for HRV, cortisol, and psychological scales.
Author Response
Reviewer 2:
This research investigated differences in stress between international and local (domestic) students. Five subjective stress scales (MBI, PSQ, PHQ-9, JSPE-S, GSE) and two physiological stress markers (HRV, hair cortisol) were used for stress measurement. The effect of nationality on JSPE-S, PHQ-9 and MBI was observed. The international students showed higher SDNN at rest, however there was no difference during seminar and during examination. Hair cortisol concentration was higher for local (domestic) students than domestic students, however the difference was not statistically significant.
Dear Reviewer, thank you very much for the effort which you have put into our manuscript. We have tried to integrate all of your suggestions and hope that the manuscript has substantially improved in this process.
1. Results on hair cortisol should be added to abstract even if there is no significant difference.
Thank you for this suggestion. We have added the following sentence to abstract’s result section (page 1, line 36):
“No effects were found for students’ hair cortisol concentrations.”
2. The authors should describe more about participant characteristics. Difficulty of studying in Germany might be different between a student from Austria and a student from Japan.
Thank you very much for this note. We have altered our Table 1 and added information about international students’ origin (page 5, line 213):
Students’ cultural background | N | Male | Female | M age | SD age |
Local | 20 | 7 | 13 | 19.80 | 2.12 |
International | 17 | 7 | 10 | 20.67 | 3.94 |
All | 37 | 14 | 23 | 20.21 | 3.10 |
International students’ origin | |||||
Europe | 6 | ||||
Southeast Asia | 4 | ||||
Latin America | 3 | ||||
Middle East | 2 | ||||
Africa | 1 | ||||
South Asia | 1 |
3. Fig. 1-4 are not necessary because the results were numerically demonstrated in the table.
Thank you for this note. However, we believe that the figures make our results more understandable to readers at a glance and would like to take a stake for keeping them, if space limitations allow. As Reviewer 1 did not recommend deleting them, how about letting the editor decide about this?
4. What was the basis for concluding that the stress of the students was low? The authors should clarify the grounds for this conclusion. Quantitative comparisons with other previous results are needed for HRV, cortisol, and psychological scales.
Thank you for this remark. It is true that we failed to provide reference values in the submitted version of the manuscript. We have now added this information to the discussion section (page 14, lines 335-339):
“In contrast to other studies [46,47], our study participants did not present significant stress levels in general – neither in the psychometrical assessments, nor in the psychophysiological or humoral evaluation – compared to the reference values of the different assessment instruments (MBI-SS [33]; PSQ [48]; PHQ-9 [38]; JSPE-S [49]; HRVs [50]; hair cortisol [51]).”
Round 2
Reviewer 1 Report
This paper could be accepted in current form.
Author Response
Dear Reviewer, thank you for your positive feedback. We highly appreciate the effort which you have invested in our manuscript.
Reviewer 2 Report
In the revised manuscript, references to the literature providing the standard value of psychological and physiological stress indicators has been added. However, specific quantitative comparisons still have not been made. The authors should describe more about the comparison with other previous results and why the authors considered that the stress of the students was low.
Author Response
Reviewer 2:
In the revised manuscript, references to the literature providing the standard value of psychological and physiological stress indicators has been added. However, specific quantitative comparisons still have not been made. The authors should describe more about the comparison with other previous results and why the authors considered that the stress of the students was low.
Dear Reviewer, thank you very much for the effort which you have put into our manuscript again.
Thank you for this remark. As you suggested, we calculated comparisons with norm samples and other previous results. In this context, we added the following information to the methods, the results as well as the discussion section:
“In case of heteroscedasticity, we used the Welch-Test to compare group differences between study participants and representative norm samples or other comparative samples.” (page 5, lines 199-200)
“3.4 Comparison between current sample and norm values
For better interpretation of our results, comparisons between the current student sample and norm groups were performed.
To the best of our knowledge, there is no MBI-SS validation study reporting mean values and standard deviations for a control sample to compare our current study results to [3]. Compared to a German students’ norm sample (M=.34, SD=.16), our study’s participants did not differ significantly (T1: M=.42, SD=.08; T2: M=.42, SD=.09; T3: M=.42, SD=.11; t(280)=4.76, p>.05) in the PSQ [36]. With respect to the PHQ-9, students in our sample presented significantly higher scores at T1 (M=4.38, SD=3.26) than a German comparative sample (M=3.56, SD=4.08; t(2097)=1.69, p<.05) .="" no="" significant="" differences="" compared="" to="" the="" comparative="" sample="" emerged="" p="">.05) at the other two points of measurement (T2: M=3.51, SD=3.20; T3: M=3.33, SD=3.22). Concerning the JSPE-S, no significant differences emerged between students in the current study (T1: M=107.3, SD=12.9; T2: M=106.2, SD=14.4; T3: M=105.8, SD=13.8) and a comparative sample of Austrian students (M=110.5, SD=12.5; t(550)=-1.99, p>.05) [47]. In the GSE, students did not differ significantly at the first two points of measurement (T1: M=30.07, SD=3.36; T2: M=30.59, SD=4.10) from a representative German sample (M=29.43, SD=5.36; t(2053)=1.69, p>.05) [42]. However, at T3, students in our study presented significantly higher values (M=32.06, SD=4.01; t(2053)=3.87, p<.05).< span="">
All students’ presented HRVs were within a normal range. Students’ RMSSD-values (resting condition: M=59.8; seminar: M=43.3; exam: M=14.7) were significantly higher than values from age-matched healthy controls (resting: M=14.2; deep-breathing: M=23.8) [48] which means they were ‘better’ and can therefore be considered as ‘normal’.
With regard to the amount of hair cortisol, students in our study had significantly higher values (T1: M=5.89, SD=5.77; T2: M=5.41, SD=4.18) than in comparable healthy samples [49].” (page 7, lines 303-326)
“In contrast to other studies [50,51], our study’s participants did not present significantly higher levels of stress with the exception of two measures: first, our sample’s students reported higher levels of depression in the PHQ-9-questionnaire at the end of the semester. However, as their scores only reach cut-off values equivalent to “minimal” depression symptoms, they should not be overrated [37]. Second, students in our sample showed significantly higher quantities of hair cortisol than comparable samples. However, we are unable to interpret these findings as our small sample size does not allow us to control for parameters, such as sex, age, and ethnicity, which may potentially affect hair cortisol measurements [49].” (page 15, lines 354-361)